# Nucleolin Targeting by N6L Inhibits Wnt/β-Catenin Pathway Activation in Pancreatic Ductal Adenocarcinoma

**DOI:** 10.3390/cancers13122986

**Published:** 2021-06-15

**Authors:** Fabio Raineri, Sandrine Bourgoin-Voillard, Mélissande Cossutta, Damien Habert, Matteo Ponzo, Claire Houppe, Benoît Vallée, Michele Boniotto, Mounira Chalabi-Dchar, Philippe Bouvet, Anne Couvelard, Jerome Cros, Anais Debesset, José L. Cohen, José Courty, Ilaria Cascone

**Affiliations:** 1University Paris Est Créteil, INSERM, IMRB, 94010 Créteil, France; fabio.raineri@inserm.fr (F.R.); sandrine.bourgoin@univ-grenoble-alpes.fr (S.B.-V.); melissande.cossutta@inserm.fr (M.C.); damien.habert@u-pec.fr (D.H.); matteo.ponzo@ircc.it (M.P.); claire.justine-houppe@inserm.fr (C.H.); b.vallee@u-pec.fr (B.V.); michele.boniotto@inserm.fr (M.B.); anais.debesset@inserm.fr (A.D.); jose.cohen@inserm.fr (J.L.C.); courty@u-pec.fr (J.C.); 2University of Grenoble Alpes, CNRS, Grenoble INP, Inserm U1055, LBFA and BEeSy, PROMETHEE Proteomic Platform, 38400 Saint-Martin d’Heres, France; 3University of Grenoble Alpes, CNRS, Grenoble INP, TIMC, PROMETHEE Proteomic Platform, 38000 Grenoble, France; 4CHU Grenoble Alpes, Institut de Biologie et de Pathologie, 38043 Grenoble, France; 5AP-HP, Groupe Hospitalo-Universitaire Chenevier Mondor, Centre d’Investigation Clinique Biotherapie, 94010 Créteil, France; 6Centre de Recherche en Cancérologie de Lyon, Cancer Cell Plasticity Department, University of Lyon, UMR INSERM 1052 CNRS 5286, Centre Léon Bérard, 69008 Lyon, France; mounira.chalabi@lyon.unicancer.fr (M.C.-D.); philippe.bouvet@ens-lyon.fr (P.B.); 7University of Lyon, Ecole Normale Supérieure de Lyon, 69342 Lyon, France; 8Département de Pathologie, Hôpital Bichat APHP DHU UNITY, 75018 Paris, France; anne.couvelard@aphp.fr (A.C.); Jerome.cros@aphp.fr (J.C.)

**Keywords:** pancreatic ductal adenocarcinoma, nucleolin, N6L, Wnt/β-catenin pathway, tumor microenvironment, cancer therapy

## Abstract

**Simple Summary:**

Pancreatic ductal adenocarcinoma (PDAC) is an aggressive cancer that has no effective treatment. Nucleolin targeting by the pseudopeptide N6L inhibits the tumor growth and metastasis of pancreatic ductal adenocarcinoma (PDAC). Here, we explored the pathways regulated by nucleolin and N6L in PDAC. We demonstrated that both interact with β-catenin, and that the Wnt/β-catenin pathway is activated in PDAC mouse models. Nucleolin inhibition decreases Wnt/β-catenin pathway activation in PDAC cells and tumors, and represents a new druggable pathway regulated by nucleolin.

**Abstract:**

Pancreatic ductal adenocarcinoma (PDAC) is a highly aggressive and resistant cancer with no available effective therapy. We have previously demonstrated that nucleolin targeting by N6L impairs tumor growth and normalizes tumor vessels in PDAC mouse models. Here, we investigated new pathways that are regulated by nucleolin in PDAC. We found that N6L and nucleolin interact with β-catenin. We found that the Wnt/β-catenin pathway is activated in PDAC and is necessary for tumor-derived 3D growth. N6L and nucleolin loss of function induced by siRNA inhibited Wnt pathway activation by preventing β-catenin stabilization in PDAC cells. N6L also inhibited the growth and the activation of the Wnt/β-catenin pathway in vivo in mice and in 3D cultures derived from MIA PaCa2 tumors. On the other hand, nucleolin overexpression increased β-catenin stabilization. In conclusion, in this study, we identified β-catenin as a new nucleolin interactor and suggest that the Wnt/β-catenin pathway could be a new target of the nucleolin antagonist N6L in PDAC.

## 1. Introduction

Pancreatic ductal adenocarcinoma (PDAC) is one of the most aggressive and lethal cancers. The incidence rate continues to increase for this type of cancer, and only 9% of PDAC patients survive beyond 5 years [1], often due to late symptom onset and diagnosis.

Nucleolin is overexpressed in several tumors and in pancreatic ductal adenocarcinoma (PDAC); 74.5% of patients have a high level of this protein’s expression, and it correlates with a poor prognosis [2,3]. Nucleolin is a nucleocytoplasmic protein involved in ribosomal assembly and mRNA metabolism, promoting cell proliferation and survival of tumor cells [4]. Our group patented the nucleolin antagonist N6L, which shows antitumor activities in breast cancer [5], prostate cancer [6], leukemia [7], non-small cell lung carcinoma [8], brain tumors [9,10], and pancreatic ductal adenocarcinoma [2,11], and furthermore can be coupled to nanoparticles [11,12,13] or toxins [10].

The Wnt/β-catenin signaling pathway has a central role in embryogenesis and is dysregulated in several types of cancers [14,15]. It enhances cell proliferation, migration, and survival. When the Wnt/β-catenin signaling pathway is not active, β-catenin is phosphorylated by casein kinase 1 (CK1) and glycogen synthase kinase 3β (GSK3β), ubiquitinated by β-transducin repeat-containing protein (β-TrCP) and degraded by the proteasome [16]. The activation of the pathway by Wnt ligands to the receptor Frizzled [17,18] induces the accumulation of β-catenin, and the transcription of target genes [19,20]. The Wnt/β-catenin pathway seems to be involved in carcinogenesis and progression of PDAC [21,22,23,24].

We are investigating molecular pathways in order to find new diagnostic markers and therapeutic targets in PDAC, in order to develop more effective therapies. N6L localizes to tumor tissue and reduces tumor growth by inducing apoptosis and inhibiting cell proliferation [2,5]. However, the mechanism of its effects on tumor cell growth was still unclear. The aim of this work was to investigate new molecular interactors of nucleolin, and we found that the Wnt/β-catenin pathway is a new pathway that is inhibited by N6L in PDAC.

## 2. Results

### 2.1. N6L Interacts with β-Catenin

The nucleolin antagonist N6L inhibits PDAC tumor growth by impairing tumor cell proliferation and survival [2]. We sought to investigate the mechanism of action of N6L by studying the N6L interactome in PDAC cells. Since N6L binds to extranuclear nucleolin, an approach of N6L pull-down of the extranuclear cellular fraction of PDAC cells and identification of proteins by mass-spectrometry was applied. Increasing concentrations of biotinylated-N6L were incubated on murine PDAC (mPDAC) cells. Cells were lysed with a specific protocol to harvest only the extranuclear fraction of cells [25], and proteins bound to biotinylated-N6L were isolated by a pull-down assay. Protein bands showed an increasing intensity from 5 μM to 10 μM in biotinylated-N6L pull-down samples, supporting a dose-dependent efficacy of the interaction with N6L (Appendix A).

Bands of proteins from the 10 μM biotinylated-N6L pull-down sample or from the pull-down of biotin alone were processed for mass-spectrometry analysis (Appendix A). Non-specific proteins found in the negative control were subtracted from the list of the specific proteins interacting with N6L. Forty-five proteins were identified as N6L partners (Appendix A). Nucleolin and 18 known interactors of nucleolin, such as integrin αvβ3 [26], belong to the list of 45 N6L partners (Appendix A). The gene ontology (GO) of the 45 proteins pulled down by N6L is built in a ClueGO network by using Cytoscape software and larger hexagons have lower *p*-value (Figure 1A). The more highly represented biological processes were the ones in which nucleolin was involved, such as “regulation of rRNA processing”, “ribosomal large and small subunit biogenesis”, “mRNA stabilization”, and “regulation of translational initiation” (Figure 1A). The same statistically overrepresented biological processes were obtained by GO enrichment analysis (Appendix A). This evidence supported the idea that this approach isolated specific N6L targets and could be used to find new N6L targets. We were interested in the biological process of “response to antineoplastic agents”, in which we identified β-catenin as a new interesting potential target of N6L. The interaction between β-catenin and N6L was validated by a pull-down assay in murine (mPDAC) and two human (MIA PaCa-2 and Panc-1) PDAC cells. β-catenin and nucleolin were present in N6L pull-down of mPDAC (Figure 1B), MIA PaCa-2 (Figure 1C), and Panc-1 (Appendix A) cells.

### 2.2. N6L Inhibits the Activation of the Wnt/β-Catenin Pathway and Prevents β-Catenin Stabilization

β-catenin is important for carcinogenesis and tumor progression of PDAC (Ram Makena et al., 2019) [27]. A N6L network (N6L interactome), integrating the 45 N6L partners and the known interactors of these partners from the STRING database, was built by using the Cytoscape software (Appendix A) and following the methods previously described and detailed in Appendix A [28,29]. We observed that β-catenin was the key node between the N6L interactome generated by the 45 proteins isolated by the N6L pull-down and Wnt/β-catenin signaling pathway (Appendix A). Pre-clinical evidence suggests that inhibition of Wnt/β-catenin signaling can impair tumor growth [15]. Since we observed that N6L interacts with β-catenin, we wondered if N6L could impact the activation of the Wnt/β-catenin pathway. mPDAC, human MIA PaCa-2, and Panc-1 cells were transfected with plasmids containing a β-catenin responding gene (*Firefly* luciferase) and a control gene (*Renilla* luciferase). Wnt3A was prepared from the conditioned media of L-Wnt-3A cells, produced, and recovered as described by Shibamoto et al. [30]. All cell lines were stimulated with Wnt3A conditioned media (Wnt3A-CM) for 24 h, and the activation of the β-catenin pathway was measured as the fold change of the luciferase activity (Figure 2A–C). Wnt3A-CM increased the luciferase activity 6.1-fold in mPDAC, 5.4-fold in MIA PaCa-2 cells, and 3-fold in Panc-1 cells, relative to controls (Figure 2A–C, respectively). N6L significantly inhibited luciferase activity 2.2-fold in Wnt3A-stimulated mPDAC murine cells relative to Wn3A-stimulated cells, but not in unstimulated cells (Figure 2A). In human PDAC cells lines, N6L impaired luciferase activity 2.16-fold and 5-fold in MIA PaCa-2 and Panc-1 cells, respectively (Figure 2B,C). In order to further understand the impact of N6L on the Wnt/β-catenin pathway, we investigated the stabilization of the protein β-catenin, which is the first event of the pathway activation. MIA PaCa-2 cells are the only human cell line, of seven different cell lines, with a low basal protein level of β-catenin, because it is constitutively degraded by the proteasome after being phosphorylated by CK1 and GSK3β [31,32,33]. The other human cell lines, including Panc-1 cells, have a high basal protein level of β-catenin [31,32,33] (Appendix A). We stimulated MIA PaCa-2 cells and Panc-1 cells with Wnt3A-CM and followed the stabilization of β-catenin protein levels by Western blotting (Figure 2D,E and Appendix A). MIA PaCa-2 cells showed a peak of an increase of β-catenin levels after 3 h of Wnt-3A stimulation, and this time point was chosen for the study of the β-catenin stabilization in MIA PaCa-2 cells (Appendix A). Panc-1 cells did not show, as expected, a clear stabilization of β-catenin under Wnt3A stimulation at different time points, probably due to the high basal level of the protein in control cells (Appendix A). The stimulation of MIA PaCa-2 cells, but not of Panc-1 cells, by Wnt3A induces a localization of the β-catenin protein into the nucleus (Appendix A). As MIA PaCa-2 cells responded to Wnt3A stimulation, we tested the effect of N6L on β-catenin stabilization in this cell line. After 3 h, Wnt-3A stimulation induced a 5-fold increase of the β-catenin levels (Figure 2D,E). The active fraction of β-catenin was studied by using a non-phosphorylated β-catenin antibody [30] (Figure 2D,F) of the non-phosphorylated β-catenin form as a marker of the correct inhibition of the β-catenin destruction machinery (APC, Axin, CK1, and GSK3β) after activation of the pathway. Non-phosphorylated β-catenin protein levels increased 5.2-fold in Wnt3A-CM MIA PaCa-2 stimulated cells compared to control cells (Figure 2D,F). The treatment of Wnt3A-stimulated cells by N6L completely inhibited the increase of the β-catenin and non-phosphorylated β-catenin protein level induced by Wnt3A-CM, but did not change the level of both forms of the protein in unstimulated cells (Figure 2D–F). The status of GSK3β activation in this setting was analyzed by using an antibody against the phosphorylation of Serine 9 that recognizes the inactive form of the kinase. The amount of phoshpo-Ser9-GSK3β was inhibited by the N6L-Wnt3A treatment compared to Wnt3A (Figure 2D,G). A GSK3β inhibitor was used to better investigate the mechanism by which N6L impacts β-catenin stabilization. GSK3β is the kinase responsible for the phosphorylation of β-catenin, causing its proteasomal destruction [16]. The treatment of MIA PaCa-2 cells for 30 min with TWS119, a GSK3β inhibitor that prevents β-catenin phosphorylation and degradation, induced a 2-fold increase in the protein’s accumulation compared to the control (Figure 2H,I). N6L abrogated β-catenin stabilization by TWS119 (Figure 2H,I). The expression of the Wnt-related target genes AXIN2 and Cyclin D1, usually followed downstream of the activation of the pathway [34], were analyzed. After 24 h of treatment, N6L significantly inhibited the accumulation of active β-catenin, and the increased expression of AXIN2 and CyclinD1 induced by Wnt3A-CM (Figure 2J–L). Together, these results suggest that N6L regulates β-catenin stabilization and functions through GSK3β.

### 2.3. Nucleolin Promotes β-Catenin Stabilization under Wnt3A Stimulation

Since N6L is an antagonist of nucleolin, and as we showed by pull-down experiments that N6L interacts with nucleolin and β-catenin, we evaluated the role of nucleolin in Wnt/β-catenin pathway activation. Nucleolin was immunoprecipitated from MIA PaCa-2 cells that were stimulated or not with Wnt3A-CM for 3 h (Figure 3A). Western blotting analysis showed that β-catenin was in the nucleolin immunoprecipitate in basal non-stimulated (−) cells, and that its amount increased after Wnt3A stimulation (Figure 3A). The interaction between β-catenin and nucleolin was confirmed in PAnc-1 cells by immunoprecipitation (Figure 3A). In order to study the role of nucleolin in the Wnt/β-catenin pathway, we performed both loss and gain of function experiments. MIA PaCa-2 cells were depleted of nucleolin by previously validated siRNAs (Gilles et al., 2016) [2], and were stimulated or not with Wnt3A. Wnt3A treatment increased the amount of the non-phosphorylated form of β-catenin in siControl cells, and to a lesser extent, in siNCL cells (Figure 3B,C). The level of non-phosphorylated β-catenin under Wnt3A stimulation in siNCL cells was about 2.8-fold lower than in the siControl cells (Figure 3C). In a gain-of-function approach, MIA PaCa-2 cells were transfected by a plasmid containing an NCL-GFP fusion gene (NCL) or a GFP plasmid (e) as control (Figure 3D,E). NCL-GFP expression increase 2.3-fold the accumulation of the non-phosphorylated form of β-catenin, compared to cells transfected with empty plasmids, under Wnt3A stimulation (Figure 3D,E). These results together suggested that nucleolin participates in Wnt/β-catenin pathway activation via β-catenin stabilization.

### 2.4. Wnt/β-Catenin Pathway Activation Specifically Sensitizes Cells to N6L Targeting

We sought to investigate the role of Wnt/β-catenin pathway activation on the viability of PDAC cells. MIA PaCa-2 cell viability was measured after stimulation with Wnt3A (Figure 4). Stimulation with Wnt3A for 24 or 72 h did not change the viability of MIA PaCa-2 cells compared to the control (Figure 4A,B, respectively), suggesting that the activation of the Wnt/β-catenin pathway alone was not sufficient to induce cell growth. Wnt3A-stimulated cells were treated by IWR-1-endo, an inhibitor of the Wnt/β-catenin pathway, or by gemcitabine, the standard of care of PDAC, or by N6L, for 24 and 72 h (Figure 4A,B). IWR-1-endo stabilizes Axin after the activation of the pathway by Wnt3A [35], maintaining the destruction complex and promoting the degradation of β-catenin. IWR-1-endo at 20 µM alone did not change the cell viability but slightly decreased the viability of cells treated with Wnt3A-CM at 24 and 72 h, compared to the treatment with IWR-1-endo alone (Figure 4A,B). N6L inhibited cell viability at 24 h and at 72 h. N6L 10 µM decreased viability by 25% and N6L 30 µM by 59.5%, respectively (Figure 4A,B). A significantly increased effect on cell viability was observed in the condition of Wnt3A-CM/N6L 10 µM compared to N6L 10 µM alone at 72 h, and in the condition of Wnt3A-CM/N6L 30 µM compared to N6L 30 µM alone, at both 24 and 72 h (Figure 4A,B). Wnt3A treatment did not improve gemcitabine sensitivity to cell viability compared to gemcitabine alone (Figure 4A,B). The synergistic effect of the combination of the treatment with Wnt3A stimulation on cell viability was calculated by using Combenefit (the High Single Agent model) [36] (Figure 4C), and IWR-1-endo showed a synergistic effect in the presence of Wnt3A stimulation, compared to treatments alone (Figure 4C). N6L, but not gemcitabine, had a significant synergistic effect in the presence of Wnt3A stimulation, and the effect increased with increasing concentrations of N6L (Figure 4C). Since N6L induces PDAC cell apoptosis and decreases cell proliferation [2], the impact of Wnt3A and N6L on MIA PaCa-2 cell apoptosis and proliferation was evaluated. N6L decreased MIA PaCa-2 cell proliferation 1.7-fold (Figure 4D) and increased cell apoptosis 4-fold (Figure 4E). Wnt3A did not impact either cell proliferation or cell apoptosis, but the combination of N6L with Wnt3A significantly increased the effect of N6L on both cellular processes by 1.6 and 2.3-fold, respectively (Figure 4D,E). These results suggested that the activation of the Wnt/β-catenin pathway could sensitize cells to the anti-proliferative and pro-apoptotic impact of N6L.

### 2.5. Wnt/β-Catenin Pathway Activation Is Essential for the Generation and Progression of MIA PaCa-2 Tumor-Derived 3D Culture

The tumor-derived 3D cultures were established from MIA PaCa-2 tumors generated in immunodeficient mice by using the complete feeding medium (CFM), enriched by many growth factors and Wnt3A-CM, as previously described in published protocols [37,38]. Tumor-derived 3D culture derived from MIA PaCa-2 tumors grown in CFM reproduced an epithelial-lined cystic structure, as previously described [37,38] (Figure 5A, Wnt3A+ arrows). To evaluate the importance of Wnt3A, we prepared a medium of culture deprived of Wnt3A-CM. Tumor-derived 3D culture derived from MIA PaCa-2 tumors and culture in the absence of Wnt3A grew slower and mostly formed small irregular masses of cells (Figure 5A, Wnt3A, arrows). This suggests that Wnt/β-catenin pathway activation is necessary for the establishment and growth of MIA PaCa-2 tumors in 3D culture, compared to the cell 2D culture of Figure 4. Tumor-derived 3D cultures grown in the presence of Wnt3A-CM were treated or not with N6L 30 µM for 24 h. They were fixed and immunostained for active β-catenin and cyclin D1 (arrows in Figure 5B,E, respectively). Tumor-derived 3D culture with Wnt3A-CM, but not treated with N6L (control), showed high immunostaining for active β-catenin (arrows in Figure 5B). Active β-catenin was mostly cytoplasmic, and no signal was evident in the nuclei of 3D cultures (Figure 5B). As it has been previously shown, endogenous β-catenin localizes to the nucleus after a short time point of Wnt3a stimulation and then nuclear staining decreases thereafter [39]. In Figure 5B, the absence of nuclear active β-catenin is probably due to persistent and non-transient Wnt3A stimulation in the growth medium, as it was observed in organoid culture of primary mouse colon tumors [40]. Nuclear Cyclin D1 staining is strong (arrows in Figure 5E). MIA PaCa-2 3D cultures treated with N6L were smaller than the controls (Figure 5C). Active β-catenin staining and nuclear cyclin D1 signals decreased after N6L treatment (Figure 5B,D–F, respectively). These results demonstrated that N6L affects Wnt/β-catenin pathway activation in 3D cultures derived from MIA PaCa-2 tumors. To support the importance of Wnt/β-catenin activation in human PDAC, we analyzed active β-catenin protein levels in 45 tumors in TMA. A total of 73.3% of the tumors had a high score (Appendix A), 17.7% an intermediate score (Appendix A), and 8.8% a low score for staining (Appendix A).

### 2.6. N6L Inhibits the Activation of the Wnt/β-Catenin Pathway and Prevents β-Catenin Stabilization

Orthotopic pancreatic tumors were induced by injecting mPDAC cells in FVB/n mice, and were treated with N6L 7 mg/kg, as previously established and described in methods [2]. Mice were sacrified and tumor volume was measured, tumor growth was inhibited by N6L (Figure 6A), as previously described [2]. Tumors show a histopathology similar to human cancer, characterized by tumoral pancreatic ducts and a hypoxic, fibrotic tumor microenvironment, with a high heterogeneity of vessel density, as in human patients [2]. The rate of proliferative cells decreased in PDAC tumors treated with N6L compared with controls, while apoptotic cells increased (Figure 6B–D), as previously described [2]. β-catenin staining was high in tumor cells of control tumors, and did not change in N6L-treated tumors (Figure 6B,E). Active β-catenin was analyzed by staining PDAC sections of control or N6L-treated mice with an antibody that recognized the non-phosphorylated form of β-catenin, as in 3D culture (Figure 6F). PDAC tumors showed a strong signal of active β-catenin specifically at the level of the tumor ductal cells (Figure 6G), suggesting that the pathway is active in vivo. N6L significantly and strikingly decreased the active β-catenin staining in tumor ductal cells (Figure 6F).

## 3. Discussion

Nucleolin overexpression in human PDAC tumors is a negative prognostic factor for patients, and nucleolin inhibition by N6L inhibits PDAC growth and metastasis in tumor-bearing mice [2]. This work aimed to investigate new N6L interactors in pancreatic cancer, to explore the mechanisms of its antitumor activity, and develop new targetable pathways in this type of cancer. We showed for the first time that N6L interacts with β-catenin, and that nucleolin inhibition impairs Wnt/β-catenin pathway activation by preventing β-catenin stabilization.

The study of the pull-down of N6L in PDAC cells allowed us to identify 45 proteins by mass-spectrometry. Nucleolin was present in the N6L pull-down, validating this approach for studying interactors of N6L. A total of 40% of the proteins of the N6L pull-down are known interactors of nucleolin, and 16% belong to the biological processes in which nucleolin is involved (regulation of rRNA processing and positive regulation of mRNA splicing via the spliceosome). We validated β-catenin, the key protein of the Wnt/β-catenin pathway, as a new interactor of N6L and nucleolin by Western blotting in different PDAC cell lines. We also demonstrated an interaction of nucleolin and β-catenin in different human PDAC cells through immunoprecipitation studies. PDAC cells were stimulated with Wnt3A, the isoform commonly used to activate the Wnt/β-catenin pathway [30,34]. The activation of the Wnt/β-catenin pathway was followed by the accumulation of the non-phosphorylated β-catenin protein, and by the transcription of Wnt/β-catenin pathway target genes. N6L prevented the stabilization of both forms of β-catenin in Wnt3A-responding PDAC cells. β-catenin is phosphorylated by CK1 and GSK3β before being ubiquitinated and successively degraded by the proteasome; the non-phosphorylated protein is, therefore, the form that is not degraded, and is considered the active form of the protein [41]. GSK3β is normally active in cells, and serine 9 phosphorylation as a negative regulation [42] was decreased by N6L in Wnt3A-stimulated cells, supporting that N6L could impact the phosphorylation of β-catenin and its degradation. In coherence with these data, AXIN2 mRNA and Cyclin D1 protein levels increased under Wnt3A stimulation, and their expression was prevented by N6L treatment in PDAC cells. In order to verify if N6L inhibited the pathway at the level of the activation of the receptor or more downstream in the signaling cascade, we used a GSK3β inhibitor, TWS119, that stabilizes β-catenin in a Wnt-independent way. TWS119 blocks the kinase activity of GSK3β, preventing the triple phosphorylation of β-catenin on serines 33 and 37 and Threonine 41, and degradation by the proteasome [39,43,44]. In this study, we showed that N6L prevents β-catenin stabilization induced by TWS119 in a Wnt ligand-independent way. This result supports that the inhibition does not occur at the receptor-ligand stage, but in the cytoplasm, downstream of GSK3β. A previous study demonstrated that nucleolin promotes Wnt/GSK3β signals to regulate stem cell signaling in hemopoiesis [45]. The loss of function of nucleolin by siRNA inhibited Wnt/β-catenin pathway activation similarly to N6L under Wnt3A stimulation. On the other hand, nucleolin overexpression increased β-catenin accumulation through Wnt3A in PDAC cells. These results suggest that the nucleolin protein level is important for the activation of the Wnt/β-catenin pathway by Wnt3A. Indeed, nucleolin depletion impairs β-catenin stabilization and nucleolin overexpression induces β-catenin accumulation, meaning that nucleolin participates in the regulation of β-catenin degradation under pathway activation.

The importance of the Wnt/β-catenin pathway is well known in tumor progression, invasion, and chemoresistance [27]. PDAC organoids from human patients have been demonstrated to be dependent to Wnt3A and R-spondin [37,46]. In this work, we have shown that the Wnt/β-catenin pathway is necessary for the establishment and growth of tumor-derived 3D culture derived from MIA PaCa-2 tumors, and strong activation of β-catenin occurs in murine models of tumors and in the MIA PaCa-2 tumor-derived 3D culture. We also observed that human cell lines cultured in 2D are not dependent on exogenous Wnt3A stimulation for proliferation and survival, and this is probably due to the conditions of culture. Hyperactive Wnt signaling is associated with aberrant signaling in colorectal cancer, and also in breast and pancreatic cancers [47]. An increase in total β-catenin and Axin2 has been observed in PDAC patients, and the Wnt pathway is one of the core signaling pathways most frequently dysregulated in PDAC [48,49,50]. Here, we showed that the majority of PDAC patients have an elevated level of β-catenin active form, supporting the idea that the Wnt/β-catenin pathway is broadly activated in PDAC. MIA PaCa-2 tumor-derived 3D cultured in the presence of Wnt3A showed a strong signal of active β-catenin concomitant to an increase of nuclear Cyclin D1 localization and accumulation. N6L treatment inhibited the growth of MIA PaCa-2 tumor-derived 3D culture and decreased active β-catenin and nuclear Cyclin D1 signals. Wnt ligands are expressed by stromal cells during pancreas tumorigenesis [46] and in the orthotopic PDAC mouse models, similarly to in PDAC patients, and the Wnt/β-catenin pathway is strongly active in ductal tumor cells. Importantly, N6L treatment is able to decrease the active β-catenin signal in tumoral ducts. In 3D culture and tumor sections from murine PDAC and patients, active β-catenin localizes mainly in the cytoplasm, as in other human tumors in which cytoplasmic β-catenin correlates with a poor prognosis [51,52].

Wnt inhibitors inhibit the Wnt/β-catenin pathway and cancer cell growth through the impairment of β-catenin stabilization [44,53,54]. Since clinical data show that elevated Wnt signaling correlates with a worse outcome for a subset of human cancers, a number of inhibitors targeting the Wnt pathway have advanced to clinical trials [47,55]. Some of these inhibitors have been shown to affect the growth of mouse models of PDAC in pre-clinical studies [56,57,58]. These inhibitors could also have an impact on the demonstrated Wnt signaling-mediated cancer therapy resistance. Nucleolin acts as a co-receptor for several growth factors and participates in different signaling pathways sustaining tumor cell growth, prompting several laboratories, including ours, to design synthetic nucleolin antagonists [4]. The nucleolin antagonist N6L selectively binds to nucleolin [5] and decreases tumor growth [2,5,7,10,13]. N6L inhibits PDAC cell proliferation, inducing cell apoptosis in mouse models and PDAC cells [2]. N6L inhibits MIA PaCa-2 cell proliferation and induced apoptosis, and this effect in 2D culture is independent of Wnt signaling because Wnt/β-catenin is not active in these cells [31]. However, the N6L anti-proliferative and pro-apoptotic effect was higher and synergic when the Wnt/β-catenin pathway was activated by Wnt3A. Cells with an active Wnt/β-catenin pathway did not respond better to gemcitabine compared to non-activated cells. This suggests that N6L acts on PDAC viability by blocking different pathways and that its effect is stronger when the Wnt/β-catenin pathway is active. This point is particularly important since we have shown that β-catenin is active in the majority of PDAC patients and because other researchers have shown Wnt/β-catenin pathway activation plays a role in drug resistance in PDAC cancer [59,60]. PDAC tumors with strongly active β-catenin could better respond to N6L treatment. Moreover, since the potential side effects of Wnt/β-catenin inhibitors are unclear, N6L could be a new potential drug for regulating β-catenin stabilization and activation.

In this work, we showed that N6L inhibits the Wnt/β-catenin signaling pathway in PDAC cell lines, tumor-derived 3D culture, and mouse models by preventing the stabilization of β-catenin. Nucleolin could promote the stabilization of β-catenin by interfering with the activity of GSK3β, which normally leads to β-catenin degradation. N6L, by inhibiting nucleolin, it may promote the degradation of β-catenin, leading to the inhibition of the Wnt/β-catenin pathway.

## 4. Materials and Methods

### 4.1. Cell Culture

Murine pancreatic cancer cells (mPDAC) were cultured as previously described (Gilles et al., 2016) [2]. MIA PaCa-2, PAnc-1, and L-Wnt-3A cells were purchased from ATCC (Manassas, VA, USA) and cultured according to the manufacturer’s instructions. Wnt3a conditioned medium (Wnt3a-CM) was produced and recovered as described by Shibamoto et al. [30], and 50% Wnt3a-CM was added to basal cell medium to activate the Wnt/β-catenin pathway.

### 4.2. Cell Growth Assay

A total of 5 × 10^3^ MIA PaCa-2 cells/well were plated in 96-multiwell plates. The day after, the cells were treated with 10 µM N6L, 30 µM N6L, 1 µM Gemcitabine (Mylan SAS France), or 20 µM IWR-1-endo (Santa Cruz, Dallas, TX, USA) in the presence or absence of Wnt3a-CM 1:2 for 24 and 72 h. Cell viability was analyzed by Alamar Blue Assay, as previously described [2]. Cell proliferation was analyzed by adding CellTrace reagents (Thermofisher, Waltham, MA, USA), following the manufacturer’s instructions, with 2 × 10^5^ MIA PaCa-2 cells/well plated in 6-multiwell plates. Cell apoptosis was analyzed by incorporating Annexin V (eBioscience, San Diego, CA, USA) with 1.5 × 10^5^ MIA PaCa-2 cells/well plated in 6-multiwell plates. To determine possible additive and synergistic effects when using combinations, the data from cell viability assays were analyzed using the freely available software, Combenefit (High Single Agent model) [36]. The software calculates a synergy score for each combination, where a positive score indicates synergy, a score of 0 is additive, and a negative score indicates antagonism.

### 4.3. RT-qPCR

1 × 10^6^ MIA PaCa-2 cells were plated in 100 mm Petri dishes and treated for 24 h with Wnt3a-CM 1:2, with or without 30 µM N6L. Cells were lysed and total RNA was purified using a PureLink RNA Mini Kit (Life Technologies, Waltham, MA, USA), according to the manufacturer’s instructions. Total RNA (100 ng) was reverse-transcribed using hexamer random primers and a first-strand cDNA synthesis kit (Fermenta, Thane, Maharashtra, India), and the synthesized cDNA was analyzed by qPCR using FastStart Universal SYBR Green Master (Roche, Basel, Switzerland). Primers used: AXIN2 (Fw: 5′GCTGACGGATGATTCCATGT3′, Rev: 5′ACTGCCCACACGATAAGGAG3′) and β-ACTIN (Fw: 5′GTTACAGGAAGTCCCTTGCCATCC3′, Rev: 5′CACCTCCCCTGTGTGGACTTGGG3′).

### 4.4. Cell Transfection

For siRNA transfection, 8 × 10^4^ MIA PaCa-2 cells/well were plated in 6-multiwell plates. The day after, cells were transfected for 72 h with 20 nM NCL siRNA (Interchim, Montluçon, France) for nucleolin or luciferase control siRNA (Qiagen, Hilden, Germany), by using Lipofectamine RNAiMAX (Invitrogen, Waltham, MA, USA). For plasmid transfection, 1.2 × 10^6^ MIA PaCa-2 cells/well were plated in 6-multiwell plates and transfected for 72 h with 5 µg NCL-GFP (nucleolin) or empty-GFP (control), provided by E.M. Reyes-Reyes (University of Arizona, Tucson, AZ, USA), using Lipofectamine 3000 (Thermo Fisher).

### 4.5. Dual-Luciferase Reporter Assay

A total of 8 × 10^5^ mPDAC, MIA PaCa-2, and Panc-1 cells/well were seeded in 6-multiwell plates. A Cignal TCF/LEF Reporter Assay Kit (Qiagen, CCS-018L), containing Firefly and Renilla plasmids, was used according to the manufacturer’s instructions, using Lipofectamine 3000 (Thermo Fisher). The activities of both Firefly and Renilla luciferase reporters were determined 72 h after transfection and 24 h after treatment with Wnt3a-CM and 30 µM N6L alone or together, using a Dual-Luciferase Assay Kit (Promega, Madison, WI, USA) according to the manufacturer’s instructions. The reporter activity is presented as the relative ratio of Firefly luciferase activity to Renilla luciferase activity.

### 4.6. Immunoprecipitation

A total of 8 × 10^6^ MIA PaCa-2 and Panc-1 cells were plated in 75 cm^2^ flasks. The day after, cells were stimulated or not with Wnt3a-CM for 3 h, and lysed using the same lysis buffer as for the pull-down. One mg of proteins of the extra-nuclear fraction was incubated with 5 µg rabbit anti-nucleolin antibody (Abcam ab22758) or 5 µg control rabbit IgG (Jackson, 011-000-002) for 1 h at 4 °C by mixing samples. Protein G sepharose beads (Millipore, Burlington, MA, USA) were prepared and added to the protein/antibody solution, according to the manufacturer’s instructions. After washing, the proteins were eluted and analyzed by Western blotting.

### 4.7. Western Blotting

A total of 4 × 10^5^ MIA PaCa-2 cells were plated in 6-multiwell plates. The day after, cells were treated for 3 or 24 h with 1:2 Wnt3a-CM, with or without 30 µM N6L. Otherwise, cells were treated for 30 min with 10 µM TWS-119 (Santa Cruz), with or without 30 µM N6L. Cells were lysed with Laemmli buffer, and protein samples were separated by SDS-PAGE and transferred onto a 0.45 μM Immobilon-P membrane (Millipore) following standard protocols. Membranes were blocked with 5% *w*/*v* BSA (Sigma, Burlington, MA, USA) in TBS 0.1% TWEEN-20 (VWR, Radnor, PA, USA) and incubated with the indicated antibodies. Immunocomplexes were visualized by using the luminescence blotting substrate ECL Pierce (Roche). Chemiluminescence was imaged by an imager system (LI-COR Odyssey Fc, Lincoln, NE, USA). Signal quantification was undertaken with Image Studio Lite (version 5.2), and expressed as the fold change relative to control. The original western blotting figures can be found in Appendix A.

### 4.8. Pancreatic Tumor Mouse Models and MIA PaCa-2 Tumour-Derived 3D Culture

mPDAC cells (1 × 10^3^ cells/mouse in 50 µL PBS) were injected orthotopically into the pancreas of FVB/n syngeneic mice, as previously described [2]. One week after cell inoculation, the mice were treated three times a week for the duration of 2 weeks by intraperitoneal injections with 7 mg/kg N6L or vehicle (saline solution) as a control. The protocol of treatment has been previously established, we defined as a starting point to perform a regression trial one week after cancer cells inoculation, a time period in which tumors reached a volume of approximately 80 mm^3^ [2]. The mice were sacrificed and total tumor size was measured as previously described [2].

MIA PaCa-2 cells (1 × 10^7^ cells/mouse in 50 µL) were injected orthotopically into the pancreas of immunodeficient mice. The mice were sacrificed after 4 weeks and tumor-derived 3D culture were established by following Tuveson’s laboratory protocol [37]. Briefly, tumors were digested with 0.012% (*w*/*v*) collagenase XI (Sigma) and 0.012% (*w*/*v*) dispase (Gibco, Waltham, MA, USA) in DMEM media containing 1% FBS (Gibco), and the resulting cell suspension was incorporated into growth factor-reduced Matrigel (Corning, Corning, NY, USA) to obtain a 3D culture.

All in vivo experiments were carried out with the approval of the appropriate ethical committee and under conditions established by the European Union.

### 4.9. Immunofluorescence and Immunohistochemistry

Tumour-derived 3D cultures established from MIA PaCa2 tumors grown in immunodeficient mice were seeded in µ-Plate 96-well black (ibidi, Gräfelfing, Germany) in 10% growth factor-reduced Matrigel (Corning). After 48 h, Tumour-derived 3D cultures were treated or not with 30 µM N6L for 24 h, were fixed in 4% paraformaldehyde (Thermo Scientific), permeabilized with 0.1% Triton-x100 (VWR), and blocked with 10% donkey serum (Jackson). Primary antibodies and secondary antibodies (Thermo Scientific D1306, 1:400) were diluted in in 0.3% BSA and incubated at 37 °C in a humid chamber for 1 h. Immunofluorescence pictures were acquired using a DSU IX81 spinning-disc confocal inverted microscope (Olympus, Tokyo, Japan) equipped with a 40× oil objective. Analysis and quantification of active β-catenin and cyclin D1 mean intensity was undertaken with ImageJ software.

Immunofluorescence of 5 µM mouse tumor slices from paraffin-embedded blocks was performed as previously described [61] by using the primary antibody for active β-catenin. The mean grey intensity of active β-catenin for each tumoral duct was measured with ImageJ, and normalized with the area of the corresponding duct.

Immunohistochemistry was performed on 5 µM mPDAC sections, dewaxed, and stained for Ki67 (Abcam ab15580, 1:500), cleaved caspase-3 (Cell Signalling, France #9661, 1:400), and β-catenin (Sigma Aldrich C2206, 1:1500). Antigen retrieval was performed with Dewax and HIER Buffer L (Thermo Scientific TA-999-DHBL) for 20 min at 95 °C. Slices were blocked with BLOXALL Endogenous Blocking Solution (VECTOR Laboratories, Burlingame, CA, USA SP-6000) for 10 min, and saturated with Blocking Solution (Zytomed, Germany ZUC007) for 5 min at room temperature. Sections were incubated with primary antibodies diluted in Antibody Diluent (Zytomed ZUC025) for 1 h, and with an ImmPRESS HRP Horse Anti-Rabbit IgG Polymer Detection Kit (VECTOR Laboratories MP-7401) for 30 min, and with ImmPACT DAB (VECTOR Laboratories SK-4105) for 10 min at room temperature. Images were acquired with a NanoZoomer S360 digital slide scanner (Hamamatsu, Japan C13220-01).

### 4.10. Statistical Analysis

Statistical analyses were performed by using GraphPad Prism (San Diego, CA, USA) (version 6). The bars represent the mean ± SEM (*n* ≥ 3). For continuous variables, we first tested both normality and equality of variance. In the case of a normal distribution, parametric statistical tests were used; otherwise, we used non-parametric tests. For two-group comparisons, we analyzed the data using the two-tailed Student *t*-test. For multiple group comparisons, 1-way ANOVA Rank with the Dunn method was used (* *p* < 0.05, ** *p* < 0.01, *** *p* < 0.005, **** *p* < 0.001).

## 5. Conclusions

This work aimed to define the mechanism of the nucleolin antagonist N6L in the treatment of pancreatic ductal adenocarcinoma (PDAC). We claim here, for the first time, that both N6L and nucleolin interact with β-catenin, and that nucleolin targeting by N6L or siRNA leads to the inhibition of the Wnt/β-catenin pathway by preventing β-catenin stabilization in human PDAC cell lines. These results have been validated in tumor-derived 3D culture and a PDAC mouse model.

## Figures and Tables

**Figure 1 cancers-13-02986-f001:**
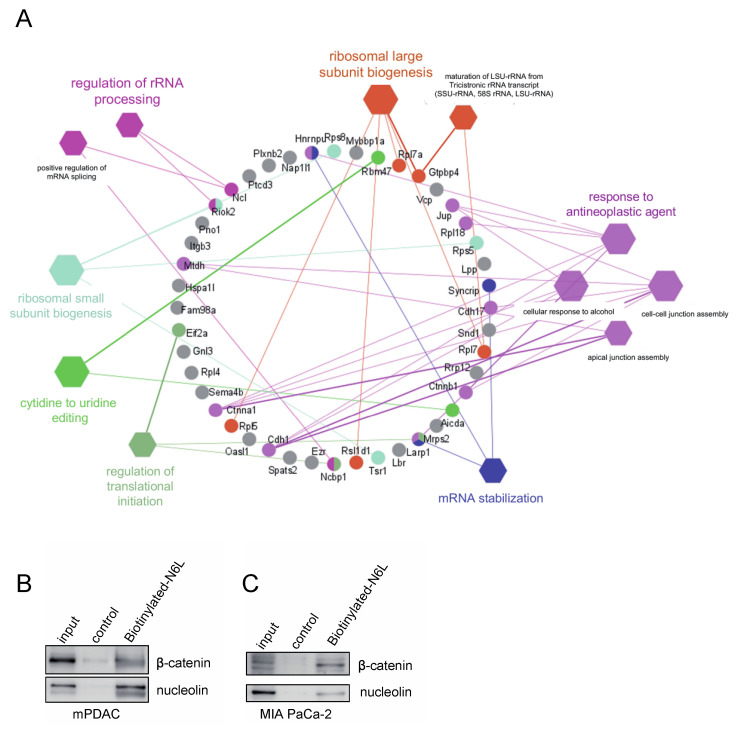
Proteins from N6L pull-down and their biological functions. (**A**) ClueGO network of non-redundant biological processes for N6L interactome (Cytoscape version 3.8.0, ClueGO version 2.5.6/CluePedia/GO_BiologicalProcess-EBI-UniProt-GOA_21.04.2020 with 21201 references). Functionally grouped network with terms (in hexagons) as nodes linked based on their kappa score level (≥0.4), where only the label of the most significant term per group is shown. The hexagon node size represents term *p*-value corrected with Bonferroni step-down level (≤0.05) (larger size for lower *p*-value). The nodes in the circle represent the genes and the edges represent the links with the biological process group. Functionally related groups partially overlap for the same gene. Gene terms not related to a functional group are shown in grey. (**B**,**C**) Western blotting of N6L pull-down. The cell lysate of the extranuclear cellular fraction (input), the biotinylated N6L pull-down and the control biotin pull-down of mPDAC and MIA PaCa-2 cells were analyzed by Western blotting with anti-nucleolin and anti-β-catenin antibodies.

**Figure 2 cancers-13-02986-f002:**
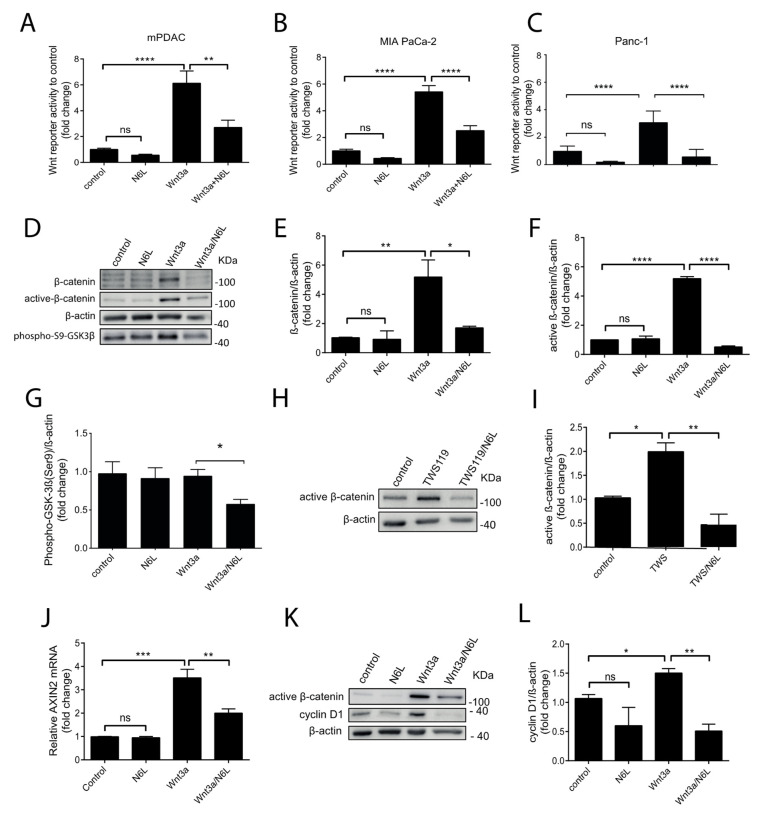
N6L inhibits β-catenin activation. (**A**–**C**) mPDACs, MIA PaCa-2 and Panc-1 cells were transfected with Firely luciferase Wnt reporter and Renilla luciferase for control and stimulated with Wnt3a-CM in presence or absence of N6L for 24 h. Graphs show the fold change of the ratio of Firefly/Renilla luciferases activity (Dual-Luciferase Reporter Assay). Wnt3A-CM induced a significant increase of luciferase activity that was inhibited in presence of 30 µM N6L in all cell lines. (One-way ANOVA, multiple comparison, *n* = 3 experiments, **** *p* < 0.001). (**D**–**G**) Western blotting of active, total β-catenin and phospho-Ser9-GSK3β of MIA PaCa-2 cells treated for 3 h with N6L and Wnt3a-CM and relative quantification are shown. Active β-catenin was analyzed by using an antibody against non-phosphorylated form of β-catenin. In (**E–G**) (One way ANOVA, multiple comparison, *n* = 5, ns = non-significant, * *p* < 0.05, ** *p* < 0.01, **** *p* < 0.0001). (**H**,**I**) Western blotting and relative quantification of active β-catenin level of MIA PaCa-2 cells treated for 30 min with 10 µM TWS119 alone or with 30 µM N6L (One-way ANOVA, multiple comparison, *n* = 3, * *p* < 0.05, ** *p* < 0.01). (**J**) qPCR analysis of AXIN2 mRNA and (**K**,**L**) Cyclin D1 protein level analysis by Western blotting after 24 h of N6L treatment in presence of Wnt3A-CM compared to Wnt3a-CM alone. (One-way ANOVA, multiple comparison, *n* = 3, ns = non-significant, * *p* < 0.05, ** *p* < 0.01, *** *p* < 0.005).

**Figure 3 cancers-13-02986-f003:**
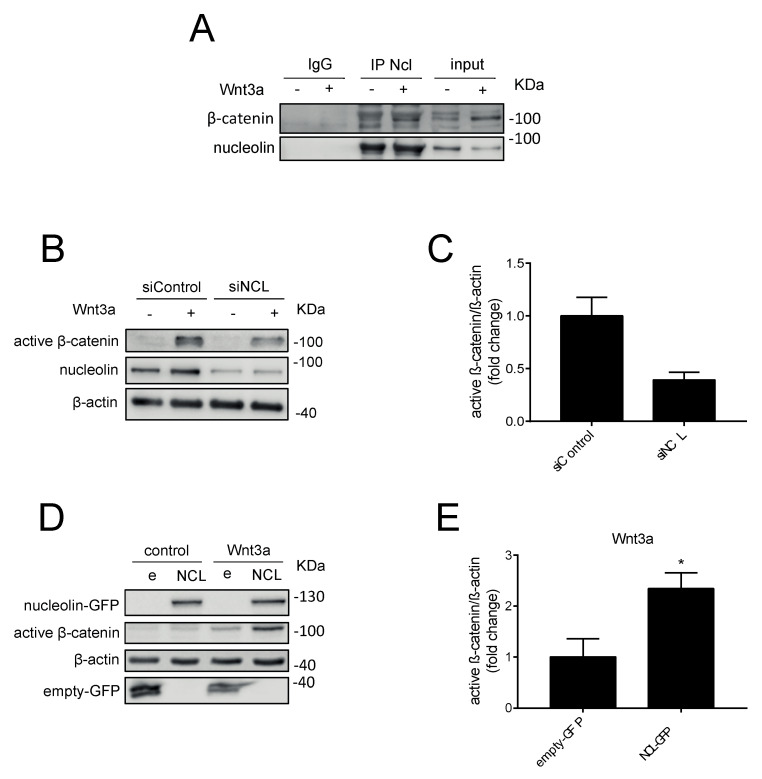
Nucleolin promotes β-catenin activation. (**A**) MIA PaCa-2 and Panc-1 cells were stimulated or not with Wnt3a-CM for 3 h. Western blotting analysis of nucleolin immunoprecipitation (IP) or control IgG performed from the extranuclear fraction is shown. Anti-nucleolin confirmed the performance of the immunoprecipitation and anti-β-catenin the interaction between these two proteins. (**B**) MIA PaCa-2 cells were transfected with 20 nM siNCL or with a control siRNA for 72 h and stimulated with or without Wnt3a-CM for 3 h before lysis. (**D**) MIA PaCa-2 cells were transfected with GFP-NCL (NCL) and GFP-empty. (e) plasmids for 72 h. Cells were stimulated with or without Wnt3a-CM for 3 h before lysis. Western blotting analysis exhibited a stronger band of active β-catenin in cells with higher nucleolin expression than those with basal levels of the protein (*n* = 3 experiments). (**C**,**E**) Western blotting quantification of active β-catenin is shown. Active β-catenin was analyzed by using an antibody against non-phosphorylated form of β-catenin (two-tailed Mann–Whitney U-test, *n* = 3 and *n* = 4 experiments, * *p* < 0.05).

**Figure 4 cancers-13-02986-f004:**
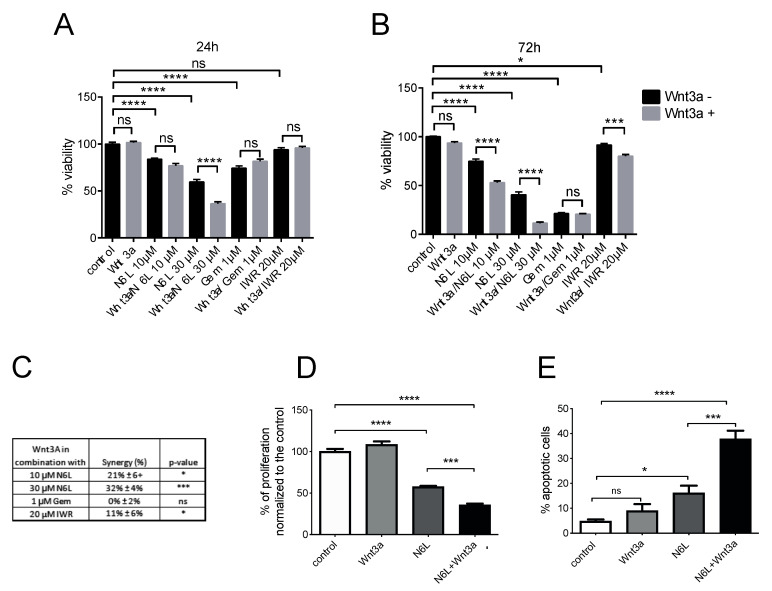
Wnt/β-catenin pathway activation specifically sensitizes cells to N6L targeting. (**A**,**B**) MIA PaCa-2 cells were treated for 24 and 72 h with 10 µM and 30 µM N6L, 1 µM gemcitabine or 20 µM IWR-1-endo in presence or absence of Wnt3a-CM. Control cells were stimulated or not with Wnt3a-CM. Cell viability was analyzed as described in Materials and Methods. (One-way ANOVA, multiple comparison, *n* = 4 experiments, ns = non-significant, * *p* < 0.05, *** *p* < 0.005, **** *p* < 0.001). (**C**) The synergy of the different combinations of treatments together with Wnt3a has been calculated by using the software Combenefit described in Materials and Methods. The score of synergy and the *p*-value are indicated in the table. (**D**) The effect of treatments on cell proliferation were evaluated by analayzing a cell tracer amount after 72 h and the graph indicates cell proliferation versus control cells. (**E**) After 24 h of treatment, apoptosis was evaluated by Annexin V staining and the graph indicates the % of apoptotic cells. (One-way ANOVA, multiple comparison, *n* = 4 experiments in (**A**,**B)**, *n* = 3 experiments in (**D**,**E**), ns = non-significant, * *p* < 0.05, *** *p* < 0.005, **** *p* < 0.001).

**Figure 5 cancers-13-02986-f005:**
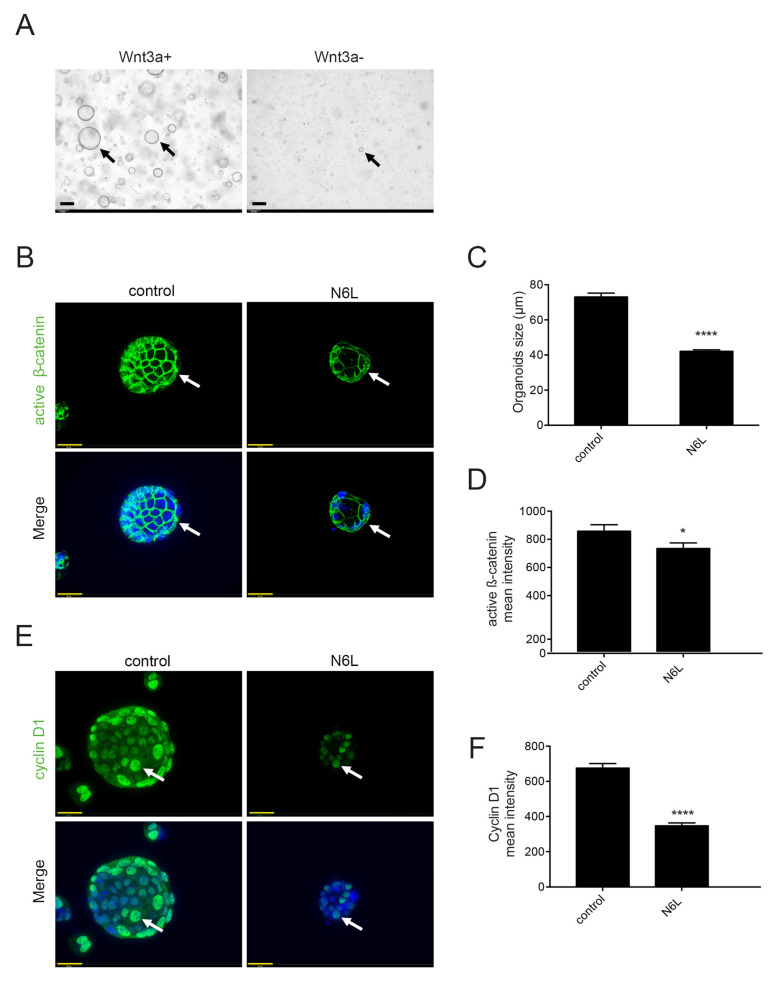
N6L inhibits Wnt/β-catenin pathway in tumor-derived PDAC organoids. (**A**) Organoid cultures derived from MIA PaCa-2 tumors (arrows) were grown in human complete feeding medium (CFM) or complete feeding medium without Wnt3a-CM. (**B**,**E**) Immunofluorescence staining by anti-active β-catenin (green) and DAPI for nuclei (blue) or by anti-Cyclin D1 (green) and DAPI for nuclei (blue) in organoid culture grown in CFM treated or not with 30 µM N6L. (**C**,**D**,**F**). Organoid size or active β-catenin signals or cyclin D1 nuclear signal in 3D culture treated or not with N6L were quantified (Two-tailed Student *t* test, *n* = 3, * *p* < 0.05, **** *p* < 0,001). Scale bars: 30 µM.

**Figure 6 cancers-13-02986-f006:**
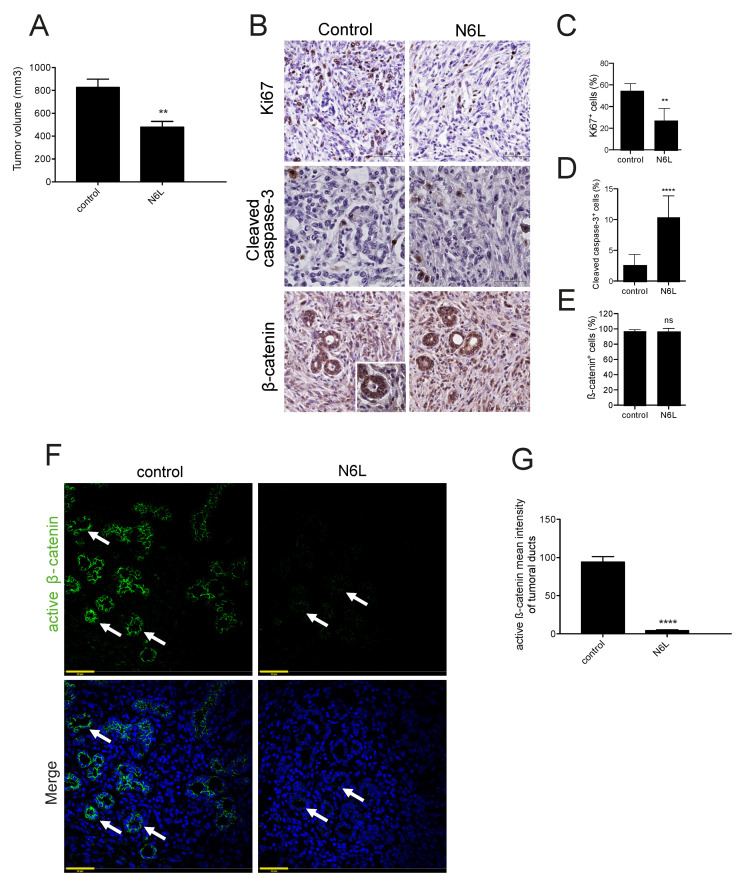
N6L inhibits Wnt/β-catenin pathway in PDAC murine model. mPDAC cells were injected into the pancreas of FVB/n mice, mice were treated with N6L and sacrificed after 3 weeks. (**A**) The volume of mPDAC tumors was measured (Student’s *t* test ** *p* < 0.01). (**B**) Sections of mPDAC tumors were immunostained by using anti-Ki67, anti-cleaved caspase 3 or anti-β-catenin. The percentage of positive cells for each staining is shown in the graph (**C–E**) (Student’s *t* test ** *p* < 0.05, **** *p* < 0.001, *n* = 4). (**F**) Sections of mPDAC tumors from mice treated with N6L or with PBS as control were stained by immunofluorescence with an anti-active β-catenin antibody (green) and DAPI for nuclei (blue). The intensity of active β-catenin staining is weaker in tumor ducts of tumors treated with N6L than the control. (**G**) Quantification of the mean grey value of active β-catenin staining in mPDAC tumoral ducts of control (*n* = 36) and N6L (*n* = 10) treated mice (Student’s *t* test **** *p* < 0.001). Scale bars: 50 µM.

## Data Availability

The data presented in this study are available on request from the corresponding author.

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
