# Peer review of "Nucleolin Targeting by N6L Inhibits Wnt/β-Catenin Pathway Activation in Pancreatic Ductal Adenocarcinoma"

_cancers, 2021, doi:10.3390/cancers13122986_

Round 1

Reviewer 1 Report

Thank you to the authors for the revised version of their manuscript. This updated version has notably improved its quality compared to the first submitted manuscript.

The authors have in general addressed my main concerns but I still have some minor comments to be considered before final publication. Some of them are minor comments not addressed by the authors in my previous review. 

  • The acronym TMO appears in the abstract section, line 57, without a previous explanation of its meaning. Please explain it the first time it appears in the text.
  • Regarding the term organoids, this reviewer appreciates the authors have now used the term TMO, as we really need to be accurate with the biological terms and models we employ in our manuscripts. MIA PaCa2 cells or tumours are not primary cells or primary murine tissue, the cells are simply injected and allowed to grow. Please ensure the term organoid does not appear in the manuscript (e.g. abstract section, line 46; line 439 methods etc).  I would recommend using something like 3D models derived from MIA Paca2 cells) or clearly specify what TMO are. 
  • In my previous review, I mentioned: "Fig 2D: labelling is missing in the last well of the WB". This has not been fixed. 
  • In vivo experiments: If the tumour volume was only measured when the animals were sacrificed, how did the authors know when to start the treatment in orthotopic models? I guess cells were injected. Authors waited for X days before starting the treatment but, how was this "starting date" selected if imaging was not performed? based on previous studies? It is not clearly explained in the text. 
  • Please delete duplicated/old figure 6, as it is confusing and I am not sure why white squares appear in Fig 6A where arrows are placed. 
  • Thank you for adding Ki67 and caspase3 studies, even if similar work was published, it is always good to include validation of current work. Perhaps authors would like to double-check the quantification performed for cleaved caspase3 IHC as the % in Fig 6D appears much higher than what Fig 6B represents. Ki67 data are more clear in the IHC studies and the quantification is not so different as in cleaved capase3. 
  • New Fig 6B b-catenin. I agree, but the WB could have been performed using both active b-catenin and total b-catenin, as I suggested in my previous review, and not only total b-catenin. Perhaps using the same antibody employed for the IF studies (Fig 6F). But thank you for the clarification. 

Reviewer 2 Report

The author significantly improved the work. Here are some minor concerns,

  1. Fig2D The 4th lane still lacks the labeling, perhaps mean the treatment of " Wnt3a/N6L".
  2. IN Fig4, the authors addressed most concerns, but combined with the generous statement from authors "In tumour, the Wnt/β-catenin pathway is activated in tumour cells by Wnt ligands expressed by stromal cells. However, some human cell lines in vitro have an autocrine activation of the pathway. To study the role of N6L and nucleolin in the Wnt/β-catenin activation we preferred use a human cell line in which the pathway is activated by adding exogenous Wnt3A as it happens in the tumour microenvrionnement. The tumour in mouse models allowed us to study the pathway activated in the tissue." It seems that in the Fig4 data, when adding extra Wnt3a alone to mimic the tumour microenvironment there is no changes in the viability (Fig4A, B), as well as no effects on proliferation (Fig4D) and apoptosis (Fig4E). How to explain this observation?
  3. Regarding the localization of active-b-catenin, the authors explained that the result of active-b-cateninstaining in organoid cultured in presence of Wnt3A, is consistent with the previous report and discussed in the discussion section. It should be better to add a little to explain why active-b-catenin is not in the nucleus in those settings. In addition, in Fig6C as mentioned by authors, in the actively growing tumors, active-b-catenin is not in the nucleus. It will be interesting to show total b-catenin staining in both Fig5B and Fig6C. 

Reviewer 3 Report

Comments on Manuscript:

  1. The authors present a study where a pseudopeptide N6L is reported to inhibit the actions of Nucleolin thereby affecting the stability of the B-catenin protein which is integral to the carcinogenesis in certain pancreatic cancer cell lines. Through various reporter assays the authors were able to shed some light on the possible mechanism of Nucleolin action.
  2. While the experimental design and execution of the study is more or less acceptable, the authors have got to do a better job in the presentation of the manuscript.
  3. The authors reference a N6L Supplementary Table 1 with the peptide/protein interaction study using a pulldown/Mass-Spec assay.  The links provided in the manuscript did not work.  Before submitting anything for review, make sure all links to any supplementary materials work.  I really would’ve liked to see some sort of an interaction affinity comparison between the 45 different protein N6L is interacting with.
  4. There are two copies of Figure 2 in the manuscript with the duplicate copy of Figure 2 containing an extra figure(2L) which is not described in the figure description.
  5. Figure 6A contains two white squares with arrows in them in the middle of the picture.  Since arrows are in the middle of the white squares I don’t know what they’re pointing to.  If this is a formatting mishap, once again please fix before submission.
  6. What should be Figure 7 is described as Figure 6 once again and the figure description below it belongs in the last figure. This is really an unacceptable formatting mistake.  It shows me a complete lack of focus when constructing this manuscript. 
  7. In page 5(of 27) of the manuscript, the authors talk about, “using a non-phosphorylated beta-catenin antibody”. I think what the authors tried to say is they used an antibody against a specific phosphorylation site of the Beta-catenin protein.  The exact residue of the protein is never mentioned, neither is the make and catalog number of that antibody(at least I couldn’t find it in materials and methods). 

Scientific Observations and recommendations:  

  1. Since the N6L peptide seems to have a lot of promiscuous interactions with many proteins, I think it would be nice to do an RNAseq study to try and capture the overall landscape of what is happening and to possibly discount greater contributions from the many protein interactions that have been reported.
  2. When doing in-vivo efficacy studies it would be nice to see some sort of tumor growth curve and maybe a Kaplan-Meyer survival curve instead just having only the weight of the tumor.
  3. It has been pretty well established that the Wnt/β-catenin pathway plays a great role in the cytoskeletal rearrangements and subsequent migration activity of a vast number of cell types including invading cancer cells and neurons. I think a lot more strength would be added to your study if you would just do simple Matrigel invasion chamber assays and some pictures of possible N6L mediated cytoskeletal architecture changes in your cell lines.

Reviewer 4 Report

In this manuscript, Rainieri et al. investigate Nucleolin as a N6L target to inhibit tumor growth and metastasis of pancreatic ductal adenocarcinoma. They explore, interestingly and in a pertinent manner, the molecular pathways regulated by nucleolin and N6L, particularly the Wnt/β-catenin pathway. They propose that both N6L and nucleolin interact with β-catenin. They also show that nucleolin targeted by N6L leads to the inhibition of the Wnt/β-catenin pathway. These results were mostly obtained in PDAC cell lines and have been validated in cell lines cultured in 3D and in a mouse model of PDAC.

This is a very interesting manuscript providing new data. Experimentally is well realized and data obtained well support their conclusions. Another important point to be highlighted is the translatability to the clinics of the result here obtained. From my point of view this is a very good work that merits to published and that will interest many scientist providing original information.

I have read the comment of reviewer 2 and 4 and in my opinion authors have conveniently answered their questions.

However, I think that the manuscript could be really improved by editing it to give more clarity. English spelling would be also edited.

I agree with reviewer 2 that the term Organoids is confusing and is still used in the manuscript. From my point of view, these are not organoids but MiaPaca2 cell growing in 3D. Another term that is confusing is “tumor” to name “MiaPaca2 xenografts”. I do not think that these semantic corrections are mandatory but they would improve the understanding of the message conveyed by this manuscript.

As a last point It was difficult to me to (as for reviewer2) understand which model was used for mass spectrometry analysis. Authors should clarify this point.

Round 2

Reviewer 3 Report

So I downloaded latest manuscript from Download Manuscript tab.

Once again, some figure errors have not been fixed.  

Figure 3 has a description of figure 3F but no actual figure 3F.

Figure 6 has two figure 6E and no description for Figure 6F.  

Unless I've been once again given the wrong version of the manuscript that's what I see. 

The authors state that it's not possible to follow tumor progression of an orthotopically transplanted tumor cell line in a mouse.  That's not true.  

You could follow orthotopic tumor progression via ultrasound:

Vevo 3100 Preclinical Imaging System | FUJIFILM VisualSonics

Or you can transfect the tumor cell line with a reporter plasmid(eg GFP or luciferase) and follow it via IVIS imaging system:

IVIS® Spectrum In Vivo Imaging System | PerkinElmer

While I do understand that the ultrasound instrument is quite rare and expensive the IVIS systems are quite common in institutes conducting in-vivo cancer research.  Contrast-CT scans can also work if you have access to that facility.

I also think that a Kaplan-Meyer curve would've been helpful as well.

Author Response

This manuscript is a resubmission of an earlier submission. The following is a list of the peer review reports and author responses from that submission.

Round 1

Reviewer 1 Report

Authors in the paper submitted to cancers titled: “Nucleolin targeting by N6L inhibits Wnt/β-­‐‑catenin pathway activation in pancreatic ductal adenocarcinoma” are presenting data on nucleolin interactions with b catenin and thus validating nucleolin as a target for PDAC therapy.

Interactions between nucleolin and Wnt/b catenin pathway are supported by interactome derived from mass spec analysis of biotinylated N6L, a specific nucleolin antagonist, and its bound extranuclear proteins. Immunoprecipitation experiments, luciferase activity with b catenin specific response elements and in vitro b catenin activation with Wnt3a-CM and specific gene expression, all of the data presented support the observation that b catenin plays a role in N6L mediated nuleolin inhibition.

Paper includes relevant data however prior publication requires several clarifications in text and in control experiments. Given that N6L was previously published as an inhibitor of PDAC growth {Gilles et al., Canc. Res. 2016}, N6L induced growth inhibition of the pancreatic cancer is not a novel idea. However detailed study on the mechanism of action such as inhibition of Wnt/B catenin pathway adds details on N6L anticancer activity.

Major concerns:

  1. In general paper needs to be edited for a proper use of English. It is recommended to have a native English speaker edit this paper.
  2. In the Abstract authors say that “N6L inhibits Wnt/b catenin pathway in vivo in murine and human tumor derived organoids” calling tumor derived organoids (TMO) in vivo experiment. However TMO is still in vitro model of PDAC growth.
  3. In Figure 2 authors analyze level of b catenin in TWS119, a GSK3b inhibitor, N6L, and Wnt3a-CM treated MiaPaCa 2 cells. Active B catenin level is decreasing with N6L treatment and combined treatment with TWS119, however authors do not include Western Blot analysis of GSK3b and its phosphorylation. GSK3b is involved in b catenin degradation, therefore it is recommended to include GSK3b protein level in this Figure. Minor critique is that Figure 2D, lane 4 is missing the full description. The level of active GSK3b would support the observation that N6L inhibitor works in b catenin pathway.
  4. In Figure 4 authors present synergy between N6L and Wnt3a-CM treatment. The experiment needs a little more explanation, since the inhibitor of b catenin pathway IWR1-endo, used as a positive control, did not provide significant growth inhibition as compared to N6L, suggesting that additional components of Wnt or B catenin pathway play a role in N6L induced growth inhibition of Wnt3a-CM treated cells. Please connect active GSK3b to growth inhibition assay, because it appears that GSK3b pathway is targeted by N6L.
  5. In verse 192 authors say synergic it needs to be corrected to synergistic.
  6. In verse 193 authors say: ”A significant additive effect on cell viability was observed”, it is not true as the effect is synergistic. The explanation of growth assay needs to be carefully explained according to the data interpretation using a provided reference for drug combination study. This part needs clarification.
  7. Discussion section is basically a repetition of Results, with simple description of listed Figures not discussion of Results in scope of current literature. It needs to be corrected accordingly, to provide explanation and relate the conclusions from experiments to previously published data on PDAC treatment with either nucleolin and/or Wnt/b catenin pathway inhibitors.
  8. In the Methods section authors describe animal experiment including both mPDAC, and human PANC-1 cells xenograft. In the results section there is clearly data from mouse PDAC cell line only.
  9. Minor corrections of presented data with either consolidation or moving parts of the results to the supplementary figures, would potentially improve the clarity of the message and increase the paper scientific impact.

Reviewer 2 Report

In this manuscript, the authors evaluate molecular pathways regulated by nucleolin in PDAC. Results show that N6L targets and inhibits the wnt/ β-catenin pathway in vitro and in vivo via β-catenin stabilization by nucleolin. This is in line with previous work published by the same authors showing that nucleolin targeting by N6L impaired tumor growth and normalized tumor vessels in PDAC mouse models. The observation included in the current manuscript is novel and potentially important. The methodology is appropriate and the conclusions, are in general, supported by the results.

There are, however, a number of points that need clarification before publication:

-Abstract lines 31-33: Please clarify in the abstract and introduction sections that the authors have previously shown nucleolin targeting by N6L impaired tumor growth and normalized tumor vessels in PDAC mouse models. It is not clear in the current version of the manuscript if this work was done by others or by the group and I think it is important to highlight the expertise of the authors in the field and clarify the novelty of the studies included in this manuscript. (e.g. we have previously shown…)

- The current version of the discussion is a bit vague and sounds like a summary of results. I will suggest improving it and highlight the possible clinical relevance and opportunities of bringing N6L into the clinic (or targeting wnt/β-catenin?). What are the main clinical limitations? Also, similar studies, if any, done in different tumor types could be included or studies supporting the idea of targeting wnt/b-catenin in PDAC. Or expand on why authors did not observe any effect on gemcitabine treated cells after wnt stimulation.

-Please avoid using the term “organoids” in this manuscript. The 3D models used here are more like spheres-like or organoids-like but not organoids per se. Only wnt was used as a growth factor and many more growth factors are needed to establish PDAC organoids following Tuveson’s protocol. Also,  “TMOs” used here are not derived from primary tissue or patient-derived material but from established cell lines. Also, control “TMOs” shown in Fig 5B-C have a size <100um. Please take these limitations into consideration.

-Intro line 51: do the authors know in what % of PDAC cases nucleolin is overexpressed? Please include this information.  And at what stage of disease? Also in PanIN lesions? Thanks

-Intro line 67: how does NL6 target specifically tumors and not normal cells?   

-In the mass spectrometry studies (section 2.1 pull-down experiments), were the 45 proteins identified as N6L partners exactly the same in mPDAC and MIA PaCa-2 cells?  Please clarify it. Why were cells treated with N6L for 45min at RT and not in an incubator? Is this based on previously published work?

-Lines 81 and 84: some concentrations are missing before uM.   Fig 2D: labeling is missing in the last well of the WB.    There is a mistake in the title of section 2.6.

- Suppl fig3A panc1 cells: please add control well to the WB, only input and biotinylated are shown. Suppl fig3B: please add b-actin and active b-catenin.

- Please explain what CM means in Wnt3A‑CM for not expert readers.

-Line 131: why did the authors only show 3 hours for MIA PaCa-2 and not a kinetic as for panc1 cells? Please show 2, 4, 8, 24h. In section 2.4 authors mention MIA PaCa cells were stimulated with Wnt for 24 and 72h…thus, this information is important.

- Line 139: the relocalization of β-catenin is not so clear based on the photos provided. did the authors use a confocal microscope for this analysis? A higher magnification could perhaps help…

-Line 208: Please briefly describe the morphology observe.  Please explain why the authors did not see differences in cell proliferation/viability in the 2D experiments (Fig 4A-B) when cells were stimulated with wnt but they observed differences using 3D models (Fig 5). 

-In vivo studies: please clarify how the authors follow the volume of the tumors if they used orthotopic models? Were the cells labeled? Indicate in the M&M section the tumor volume when the treatment was started. Please include in the main text the concentration of N6L used in these experiments also if it is cited in the M&M.

-Figure 6B: it is a too general conclusion only based on tumor volume post-mortem (I guess?). An IHC for active caspase-3 or Ki67 should have been included to validate the decrease in tumor volume and better understand if this was due to a stop in proliferation or cell killing. Is there any damage to the normal pancreas? These data are not shown…Do authors have H&E staining post-treatment?

- Fig 6C: validation, for example by WB, of active and total b-catenin should be included

-Why did the authors do not include any in vivo results using MIA PaCa-2 cells to validate the results observed in 2D and 3D models?

Reviewer 3 Report

Fabio Raineri et al here described the N6L, a potential nucleolin antagonist, inhibit b-catenin activation. The introduction provided little information on N6L, hence the rationale of this work is less clear. They provided several biochemical and cellular data on the mechanisms that N6L interacts with nucleolin and b-catenin and destabilizes b-catenin to inhibit the wnt/b-catenin pathway and its functions. The key question is that how do N6L, nucleolin and b-catenin interact with each other, there is no evidence that nucleolin is the target of N6L.  potential mutational variants of nucleolin and b-catenin, N6L are required for the IP assay, in addition, the confocal microscopy of the cellular localization of N6L,b-catenin, and nucleolin should also be required. The author stated that Wnt/β-­catenin signaling pathway is active in several cancers including PDAC, such as Fig5 data, so when they choose the PDAC cell lines for cell assay, why do they choose the cell lines without the active Wnt/β-­catenin and need the addition of Wnt3A to activate the Wnt/β-­catenin in mPDACs, MIA PaCa-2 and Panc-1 cells.

1.Fig2D, one lane on the right misses the labeling.

2.Fig3A, legend"The band of the IP of cells stimulated with Wnt3a appeared more intense than the band of cells not stimulated (n=3 experiments)." actually there is hardly changes between the -+Wnt3a treatment. there is no Fig3b figure legend description.

3.Figure legends, "in the presence or not of " should be "in the presence or absence of ".

4. In Fig4, How does N6L treatment decrease cell viability, while it seems that there is no evidence that N6L decreases the basal b-catenin levels showed in Fig 2D? Also in the presence of wnt3a, it increases the b-catenin levels but no effects of it on the viability. The WB data on b-catenin should be included.

5.In  Fig5b, why active b-catenin is not locatedin the nucleus? the N6L, nucleolin, and catenin and cellualr compartment markers shoud be co-stained here.

Reviewer 4 Report

In this manuscript, Rainieri et al. investigated the mechanisms underlying the anti-tumor effect of nucleolin antagonist, N6L, in pancreatic cancer. They propose N6L prevents the stabilization of beta-catenin, thereby inhibiting tumor cell survival.

My major concern is the data are not convincing to reach the conclusion that “endogenous” beta-catenin is a significant player that mediates the anti-tumor effect of N6L. In the majority of the figures (Figs 2, 3, and 5), exogenous Wnt3a was supplied, which were used as the system to generate most of the mechanistic insights. However, In Figure 4, without exogenous Wnt3a, N6L was sufficient to kill tumor cells, suggesting that something else is at play. Granted that when supplied “extra” Wnt, N6L degrades beta-catenin, but the scenario of extra Wnt may not be relevant.

A second major concern is along the same line. The focus of the manuscript argues that N6L-nucleolin reaction inhibits tumor cell viability through degradation of beta-catenin, as a major mechanism underlying the anti-tumor effect of N6L. However, the observations that Wnt3a did not change cell viability despite upregulating active beta-catenin suggests that this proposed mechanism may not be significant.

Several minor concerns are as follows:

The authors’ data do not support their statement that GSK3b regulates N6L-mediated beta-catenin stability: such conclusion can only be reached by comparing beta-catenin levels upon N6L treatment vs. N6L + TWS119 treatment.

What is Wnt3a-CM? Conditioned media? How was this produced?

The authors drew incorrect conclusions on “cell growth” based on Amalar viability assay.

Along the same line, N6L inhibition could be a result of increased cell death or reduced tumor proliferation, or a combination of two. An alternative assay should be used to address these possibilities.

In the final figure it is weird that tumor volume (usually used to monitor the growth of subQ tumors with time) was used to measure orthotopic PDAC. Please show end point wet weight measurement.

Not enough background was given on the relationship between nucleolin and Wnt in tumor cells.
